# HS-GC-IMS and PCA to Characterize the Volatile Flavor Compounds in Three Sweet Cherry Cultivars and Their Wines in China

**DOI:** 10.3390/molecules27249056

**Published:** 2022-12-19

**Authors:** Baoxiang Liu, Yang Yang, Likun Ren, Zhengbo Su, Xin Bian, Jing Fan, Yuanyuan Wang, Bing Han, Na Zhang

**Affiliations:** 1College of Food Engineering, Harbin University of Commerce, Harbin 150076, China; 2Shandong Provincial Key Laboratory of Food and Fermentation Engineering, Shandong Food Ferment Industry Research & Design Institute, Qilu University of Technology (Shandong Academy of Sciences), Jinan 250013, China

**Keywords:** cherry, cherry wine, volatile compound, HS-GC-IMS, PCA

## Abstract

The aim of this research was to characterize differences and sources of volatile flavor compounds by using headspace-gas chromatography-ion mobility spectrometry (HS-GC-IMS) and principal component analysis (PCA). Three sweet cherry fruits from different cultivars (cv. Tie, Van, and Lap) and their wines that were produced by the same yeast were detected. The results showed that 27 flavor compounds were identified in cherry fruits, including 10 alcohols, 7 esters, 7 aldehydes, 2 ketones, and 1 organic acid. Twenty-three flavor compounds were identified in cherry wines, including nine esters, eight alcohols, three aldehydes, two organic acids, and one ketone. In cherry fruits, aldehydes, several alcohols, and one ketone were the most prevalent in cv. Tie, and the majority of esters and alcohols in cv. Van. After fermentation, ethanol, butanol, butanal, ethyl propionate, propionaldehyde, 3-hydroxy-2-butanone, and acetic acid increased, whereas 1-hexanol, 3-methyl-3-buten-1-ol, 1-penten-3-ol, ethyl acetate, methyl acetate, *(E)*-2-hexenal and hexanal decreased. Few differences were detected in the type and content of volatile compounds in cherry wines from cv. Tieton (WT) and cv. Van (WV). Almost all aldehydes are derived from cherry fruits, which cannot be produced during wine-making, and other volatile compounds are almost all produced by *saccharomyces cerevisiae*. The volatile compounds of cherry wines were determined by row materials and fermentation cultures. Flavor fingerprints were established by HS-GC-IMS and PCA, which provided a theoretical foundation for the evaluation and improvement of flavor quality in cherry wine-making.

## 1. Introduction

Sweet cherry (*Prunus avium* L.) is popular and well favorite by consumers due to its bright color, particular odor, pleasant taste, and nutritional ingredients [1]. It contains high nutrients and bioactive compounds, such as carbohydrates, amino acids, vitamins, phenolics, organic acids, flavonoids, anthocyanidin, and mineral substances, among which cherry flavonoids show high antioxidant capacity [2,3,4,5,6]. Furthermore, cherries could promote the endogenous production of melatonin and establish high-quality sleep [7]. However, due to its high moisture and sugar content, cherry is a very perishable fruit, which is usually processed into juices and cherry wines to ensure the supply of cherry derivatives throughout the year [8]. Different production processes of fruit wines will also lead to different volatile components. For example, fermentation is a traditional food processing technology that could extend preserving shelf life [9], and it could also promote quality through sensory features and potential health-promoting effects [10].

Flavor is a crucial sensory property of fruits and their fermented drinks, which is still the primary selection criteria for consumers. The cultivars of raw materials may have a significant influence on fruit wines, including sensory properties. Physicochemical characteristics such as weight, dimensions, color, pH, texture, soluble solids content (SSC), and titratable acidity of different cherry cultivars may affect food quality and sensory properties [11,12].

Thereinto, aldehydes, alcohols, and esters were the key volatile compounds in cherry fruits [13,14]. It was found that *(E)*-2-hexenal, hexanal, benzaldehyde, ethanol, 1-hexanol, benzyl alcohol, ethyl acetate, and ethyl caproate were the key volatile compounds [15,16]. Aldehydes existed throughout the growth process; the contents increased quickly in the color stage and decreased in ripening as to the phytohormone [17]. In contrast, the content of alcohols and esters is raised during the maturation of cherry [18]. The volatile compounds were detected by using gas chromatography-mass spectrometry (GC-MS), and it was found that alcohols and esters were the primary volatile group in cherry wines [19,20,21]. However, there are few reports about differences and sources of volatile flavor compounds of different cultivars, and the source mechanism is still not clear in cherry wines.

This far, there are several ways to measure volatile compounds by using various techniques to enhance and distinguish the characteristics of cherry wines, such as GC-MS, GC-O (Gas chromatography-olfactometry), and coupled with other means [16,19,20]. GC-MS is the most applied tool to differentiate cherries and wines in fruit development, raw materials, cultures, and processing methods [18,19,20,21]. GC-IMS could separate volatile compounds in two dimensions with high separation efficiency with less time and cost compared with GC-MS [22]. IMS could operate without high pressure, and HS means that the samples need no pretreatment. With the advantage of less analysis time, a lower limit of detectability, good separability, and repeatability, the HS-GC-IMS technique could be a powerful way to analyze volatile compounds in multiple samples [23,24,25]. PCA is a key technique to distinguish the differences between multiple samples by summarizing into much fewer variables which are a weighted average of the original variables. The data mining technique and multivariate linear transformation are used to expound the main contributors and analyze the differences in aroma, sensory, and taste properties [26,27,28].

In this research, the volatile flavor of three cherry fruits and their wines were detected. An effective flavor evaluation method was established to characterize the difference and source of volatile flavor compounds in cherry wines by HS-GC-IMS and PCA. As exploratory research, this work may promote the application and quality improvement for the making of cherry alcohol beverages.

## 2. Materials and Methods

### 2.1. Materials

In this research, cherry fruits of three different cultivars were collected (May 2021) from Ai Ying Wei ecological manor in Rizhao city (Shandong, China). Commercial active dry wine yeast BV 818 was purchased from Angel Yeast Co., Ltd. (Yichang, China) and Pectinex Ultra SP-L from Novozymes (1000 U/g, China) Biotechnology Co., Ltd. (Tianjin, China). Sucrose was purchased from Rizhao Lingyunhai Sugar Co., Ltd. (Rizhao, China).

### 2.2. Fermentation Process of Cherry Wines

Cherry fruits of different cultivars were selected as the main material. Cherry was washed, manually removed the peduncle, destemmed, and crushed. Pectin was degraded by pectinase (0.5 mL/L), then about 13% of sucrose (m/m) was added to the cherry pulp. The SSC could attain 25 (°Brix). Commercial yeast (300 mg/L) was inoculated for fermentation promoter, and alcohol fermentation process was implemented at 23 ± 1 °C for about 160 h throughout static fermentation. During fermentation, every fermented broth was stirred twice or thrice a day to keep the pulp uniformity. After fermentation, all resulting wines were separated with centrifugalization at 2808× *g* for 10 min (room temperature) to terminate the fermentation.

### 2.3. HS-GC-IMS Analytical Conditions

Analyses of volatile compounds were performed with an automatic HS sampling unit (CTC Analytics AG, Zwingen, Switzerland). GC unit (Agilent 490, Agilent Technologies, Palo Alto, CA, USA) and IMS instrument (FlavourSpec^®^, Gesellschaft für Analytische Sensorsysteme mbH, Dortmund, Germany) were combined. The conditions of the automatic sampler and GC-IMS are shown in Table 1.

### 2.4. Statistical Analysis

The identification of the volatile compound was according to the retention index (RI) and drift time (RIP relative) in the GC-IMS library. The spectrogram was analyzed by its accompanied software, including VOCal and three plugins (Reporter, Gallery Plot, and Dynamic PCA plugins). By using the topographic plots, Reporter plugin can analyze the spectral differences in different samples. The fingerprint comparison was shown by the Gallery plot plugin of cherry fruits and their wines derived from three cultivars. The data were processed with the peak intensity of detected volatile compounds by using the dynamic PCA plugin. The PCA data matrix of cherry fruits had 42 columns (names of volatile compounds) and nine rows (three samples with three replicates). In cherry wines, there were 35 columns and nine rows. Then, the peak intensity was scaled and mapped to appropriate intervals from the matrix automatically. The principal components were determined, and the corresponding contribution rate calculations were performed. Finally, scores and loading plots were drawn to detect the regularity and difference between cherry fruits and their wines. All the samples were tested in triplicate.

## 3. Results and Discussion

### 3.1. Characterization of Sweet Cherry Fruit from Three Cultivars

As shown in Figure 1, volatile compounds intuitively differ in three sweet cherry cultivars. The three-dimensional (3D) respectively represented the ion drift time, gas phase retention time, and peak intensity. Each peak signal represented one type of volatile compound. The results demonstrated visualized differences in different cultivars, which also inferred differences in their wines.

GC-IMS detection results were shown in a pseudo-color map (Figure 2) to differentiate the topographic plot of cherry fruits from three cultivars intuitively. In the two-dimensional (2D) topographic map of different cultivars, X and Y axis meant the drift time and retention time, respectively. Each point represented one type of volatile compound or a dimer in the spectrum. The grayscale image represented the normalized difference. The red or blue point indicated that the signal was strong or weak, which meant that the relative content was higher or lower. The integration parameters of volatile compounds were detected in Table 2. Moreover, it could be observed that some flavor compounds appeared as proton-bound monomers and dimers.

A total of thirty-six signal peaks were identified in cherry fruits, corresponding with twenty-seven volatile compounds, including ten alcohols, seven esters, seven aldehydes, two ketones, and one organic acid. Most of the signals were concentrated in 200–1000 s (retention time) in cherry fruits. Compared with cv. Tie and Lap, cv. Van had more red points. It was demonstrated that cv. Van had the most abundant content of volatile compounds.

Benzaldehyde, *(E)*-2-hexenal, hexanal, and heptanal showed the most abundant signals of volatile compounds in cv. Tie, and alcohols, esters, and organic acid showed the most abundant signals in cv. Van. There were only 2-methyl-1-butanol, ethyl propionate, and trans-2-methyl-2-butenal showed abundant in cv. Lap. Alcohols may derive from the infection by wild yeast when the cherry was on the trees [29], which is also a source of cultures from spontaneous fermentation. Alcohols possibly promoted the generation of esters, whereas they decreased aldehydes.

Figure 2 and Table 2 show the main volatile compounds of different cultivars. Aldehydes such as benzaldehyde, *(E)*-2-hexenal, hexanal, trans-2-methyl-2-butenal, butanal and propionaldehyde, esters such as ethyl acetate, hexyl acetate, isoamyl acetate, and methyl acetate, alcohols such as ethanol, 1-hexanol, and 1-penten-3-ol, and an organic acid such as acetic acid were abundant in all three cultivars. The results correspond to some researchers’ reports that benzaldehyde, *(E)*-2-hexenal, hexanal, ethanol, 1-hexanol, and ethyl acetate were key volatile compounds in cherry fruits [15,16].

The fingerprint analysis was given to qualitatively characterize the volatile compounds in different cherry cultivars (Figure 3). In the fingerprint, every row represents one kind of sample, and every column represents one kind of volatile compound. The light of each square roughly represents the content of each volatile compound. As described above, the aldehydes were the most prevalent in cv. Tie, for example, benzaldehyde, *(E)*-2-hexenal, hexanal, and heptanal. Furthermore, 3-methyl-3-buten-1-ol, 1-penten-3-ol, butan-2-ol, and 4-methyl-2-pentanone also showed the most abundant. Almost all esters and alcohols showed the most prevalent in cv. Van. Except for trans-2-methyl-2-butenal and ethyl propionate, there was no volatile compound that had more prevalent in cv. Lap.

The scores plot of PCA in three cherry cultivars was shown in Figure 4, the accumulative variance contribution rate was 73% and 21%, and the total cumulative contribution ratio exceeded 60%, so the PCA model can be regarded as a well-separated instrument [27]. These components showed good repeatability in cherry fruits of different cultivars. There was a significantly different composition and content of volatile compounds of Tie, Van and Lap. The main contributing volatile compounds of cv. Tie to the differentiation were 4-methyl-2-pentanone heptanal, (E)-2-hexenal, 3-methyl-3-buten-1-ol, benzaldehyde, butan-2-ol, ethyl acetate, methyl acetate (Appendix A). 2-methyl-1-propanol, trans-2-methyl-2-butenal, ethyl propionate, 3-methyl-1-butanol and acetic acid contributed to cv. Lap. Hexyl acetate, isoamyl acetate, 3-methyl-1-butanol, 2-methyl-1-propanol, 3-hydroxy-2-butanone, propionaldehyde, isobutyl acetate, 1-hexanol, and butanal contributed to cv. Van. Hence compared with cv. Lap, the odor type of cv. Tie were fruity and green, and cv. Van were fruity and ethereal. Combing with the results of Figure 3 and Figure 4, HS-GC-IMS and PCA could be ways to distinguish different cherry cultivars.

### 3.2. Characterization of Cherry Wines Derived from Three Cultivars

As shown in Figure 5 and Figure 6, there were a few differences in WT and WV, and the contents of some volatile compounds were higher or lower in the cherry wine (WL) from cv. Lapins. In primary fermentation, more alcohols and esters were produced by yeast, which constituted the dominating volatile flavor compounds in cherry wines. Combined with Table 2, twenty-nine signal peaks were identified in cherry wines derived from different cultivars, corresponding with twenty-three volatile compounds. Among them, twenty-three compounds were composed of eight alcohols, nine esters, three aldehydes, two organic acids, and one ketone.

There were few differences in the content of alcohols (1-hexanol, 3-methyl-1-butanol, 3-methyl-3-buten-1-ol, 1-propanol, and ethanol) of all three samples. The content of alcohols may be determined by fermentation cultures, which corresponded to some research [30]. Except for ethanol, higher alcohols such as 3-methyl-1-butanol, 2-methyl-1-propanol, 1-hexanol, and butanol might be generated through glycolysis and the Ehrlich pathway in the yeast [31]. The content of higher alcohols is mainly relative to ferment factors and the content of amino acids [32], which means that the three cherry samples shared a similar constitution of amino acids. Moreover, WT and WV hold the similar type and content of some volatile elements, including isoamyl acetate, ethyl propionate, propyl acetate, ethyl acetate, butanal, 2-methylbutanal, propionaldehyde, and propionic acid, as shown in the Gallery Plot (Figure 7). Compared with WT and WV, isoamyl acetate, isobutyl acetate, ethyl propionate, ethyl hexanoate, ethyl acetate, and methyl acetate showed relatively high content in WL, whereas two esters (propyl acetate, butyl acetate), all aldehydes, organic acids and ketone showed the relatively lower content. The content of ethyl octanoate, ethyl hexanoate, and methyl acetate in WT was higher than in WV, while isobutyl acetate, butyl acetate, and 3-hydroxy-2-butanone were lower.

As can be seen from Figure 8, the accumulative variance contribution rate was 89% and 9%, and the total cumulative contribution ratio was 98%. The scores plot of PCA showed good repeatability in cherry wines derived from different cultivars. The main contributing volatile compounds of WT to the differentiation were propyl acetate, 1-penten-3-ol, 2-methyl-1-propanol, and 1-hexanol (Appendix A). Isobutyl acetate, isoamyl acetate, ethyl propionate, ethyl acetate, methyl acetate, ethyl hexanoate, and ethyl octanoate contributed to WL. Butanol, butyl acetate, 1-propanol, butanol, and acetic acid contributed to WV. Those correspond to the result of the Gallery Plot. There were more differences in volatile compounds in WL, which could be distinguished by PC1. Combing with PC2, PCA could apparently separate those cherry wines of different cultivars. Thus, cherry fruits and their wines derived from different cultivars could be distinguished by the Gallery Plot and PCA.

On the whole, the main volatile compounds of cherry wines were alcohols and esters. Esters were more abundant in WL compared with WT and WV. The odor type of WT and WV was largely determined by alcohols, while WL was by esters. Hence the odor of volatile compounds was fruity and ethereal in WL. The volatile compounds might be largely determined by the type of fermentation cultures in cherry wines.

### 3.3. The Source of Volatile Compounds in Cherry Wines

From the perspective of types of volatile compounds, alcohols and esters generally existed in cherry wines. A number of esters were produced by substrate esterification, yeast enzymatic metabolism, and the metabolism of aliphatic acid during fermentation [33]. The main esters were acetate esters, such as isoamyl acetate, isobutyl acetate, propyl acetate, ethyl acetate, methyl acetate, and butyl acetate, which significantly influenced flavor and fragrance. There were several alcohol-O-acetyl (or acyl) transferases (AATases) in yeast; thereinto, Atf1 and Atf2 were in charge of producing acetate esters in *Saccharomyces cerevisiae* [34]. *S. cerevisiae* can produce volatile compounds (like esters) through de novo synthesis or biotransformation from the appropriate substrate. The content of acetate esters is usually affected by the existence of higher alcohols and fatty acids during fermentation [35]. Esters were generally regarded as the principal factor of the flavor of alcoholic beverages, bringing about fruity and ethereal odor.

Higher alcohols had a positive-going effect on flavor despite their higher sensory threshold when they held appropriate content [36]. The content of 1-hexanol, 3-methyl-3-buten-1-ol, 2-methyl-1-butanol, and 1-penten-3-ol decreased after fermentation, whereas they showed relatively high content in all three cherry fruits. Furthermore, two esters (hexyl acetate and methyl acetate) were derived from cherry fruit. This meant that these compounds were not produced by yeast. Those alcohols may derive from the maturation process of cherry and infection of wild yeast [18,29]. As to the metabolism of commercial *Saccharomyces cerevisiae* (BV818), 3-methyl-1-butanol, butanol, 2-methyl-1-propanol, and 1-propanol noteworthily increased and showed relatively high content in all cherry wine samples.

Alcohols may promote the production of esters, whereas decreased aldehydes. The majority of aldehydes, such as hexanal, nonanal, trans-2-methyl-2-butenal, and heptanal, decreased due to the volatilization with fermented CO_2_ and oxidation that converts aldehydes into their respective acids during fermentation. As the main volatile compounds, aldehydes existed throughout the growth process, while their contents increased quickly in the color stage and decreased in maturation [18]. Thus, those aldehydes derived from cherry fruits cannot be produced during fermentation. Conversely, 2-methylbutanal and propionaldehyde showed low content in cherry fruits and significantly increased in cherry wines, bringing fusel and ethereal aroma. The increasing content of aldehydes may be derived from oxidization during crushing, enzymolysis, and fermentation processes, which may degrade some lipids into aldehydes [5,37]. Meanwhile, the yeast produces a mass of propionic acid, acetic acid, and 3-hydroxy-2-butanone, bringing with acidic and buttery odor. The other volatile compounds are almost produced by *saccharomyces cerevisiae*.

Some volatile compounds, such as isobutyl acetate, increased in WT and WL but decreased in WV, and the butanal increased in WT and decreased in WV and WL. The starter culture of yeast could regulate the content of some volatile flavor compounds, making cherry wines affected less by raw materials. Therefore, the volatile compounds of cherry wines were regulated and affected by fermentation microorganisms.

## 4. Conclusions

In conclusion, aldehydes, alcohols, and esters were prevalent in cherry fruits, and huge diversity in the content of aldehydes, while esters and alcohols were ubiquitous in wines. Distinct differences in volatile compounds could be ways to distinguish different cherry fruits and their wines. Sources of some volatile compounds in cherry wines were from cherry fruits (almost aldehydes, four alcohols and two esters), and the content were determined by cherry cultivars. The content of alcohols showed few differences in three cherry wines. Cherry fruit (Tie) showed relatively low content of esters, but its wine (WT) was similar to WV. This meant that volatile compounds of cherry wines might be largely determined by fermentation cultures. This research may provide a theory foundation for the evaluation and improvement of flavor quality in cherry wines making.

## Figures and Tables

**Figure 1 molecules-27-09056-f001:**
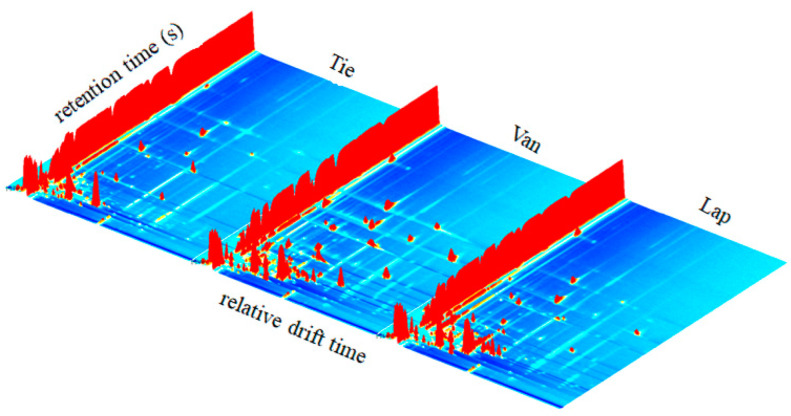
Differences in the three-dimensional (3D)-topographic of volatile compounds in cherry fruits from three cultivars (Tie: cherry cultivar Tieton; Van: cherry cultivar Van; Lap: cherry cultivar Lapins).

**Figure 2 molecules-27-09056-f002:**
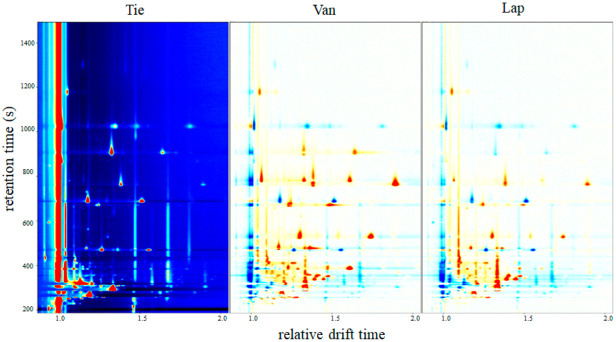
Differences in the two-dimensional (2D)-topographic of volatile compounds in cherry fruits from three cultivars.

**Figure 3 molecules-27-09056-f003:**
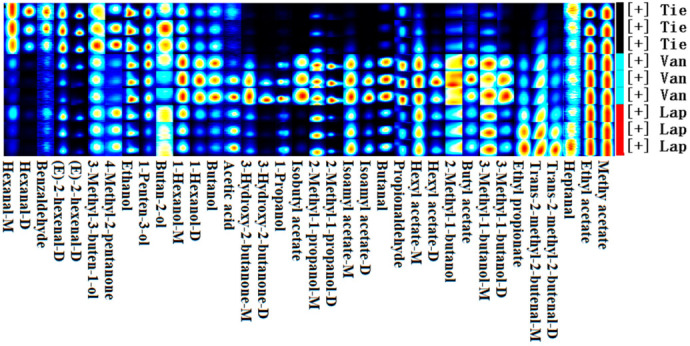
The Gallery Plot of volatile compounds in cherry fruits from three cultivars.

**Figure 4 molecules-27-09056-f004:**
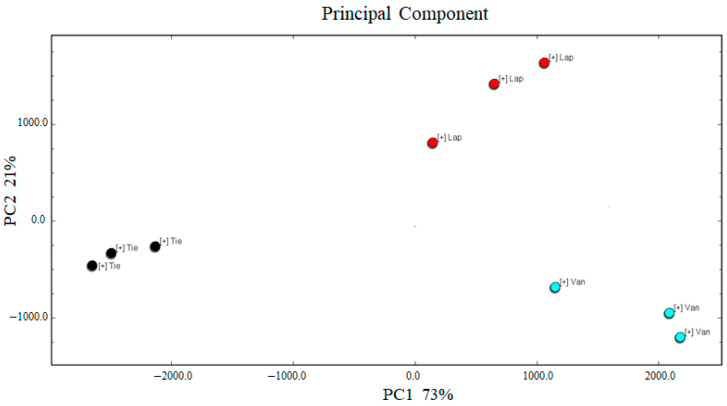
The scores plot of PCA analysis in cherry fruits from three cultivars.

**Figure 5 molecules-27-09056-f005:**
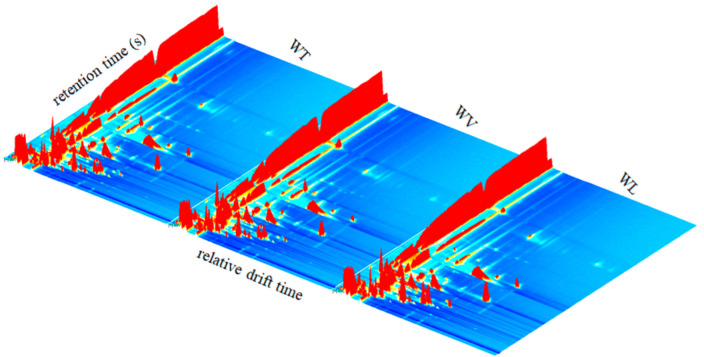
Differences in the three-dimensional (3D)-topographic of volatile compounds in cherry wines derived from three cultivars (WT: cherry wine from cultivar Tieton; WV: cherry wine from cultivar Van; WL: cherry wine from cultivar Lapins).

**Figure 6 molecules-27-09056-f006:**
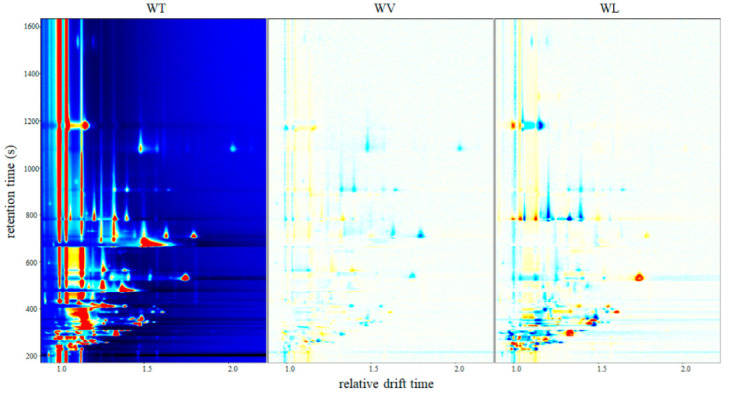
Differences in the two-dimensional (2D)-topographic of volatile compounds in cherry wines derived from three cultivars.

**Figure 7 molecules-27-09056-f007:**
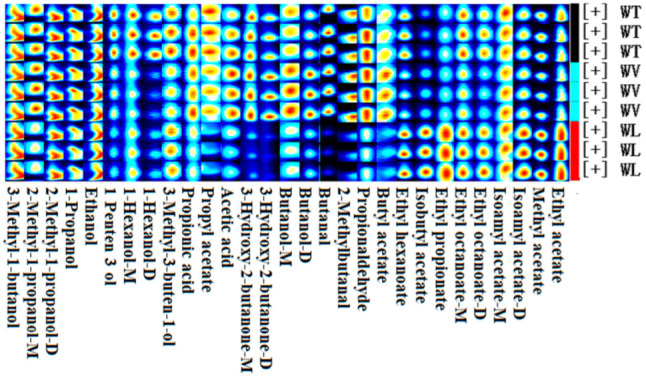
The Gallery Plot of volatile compounds in cherry wines derived from three cultivars.

**Figure 8 molecules-27-09056-f008:**
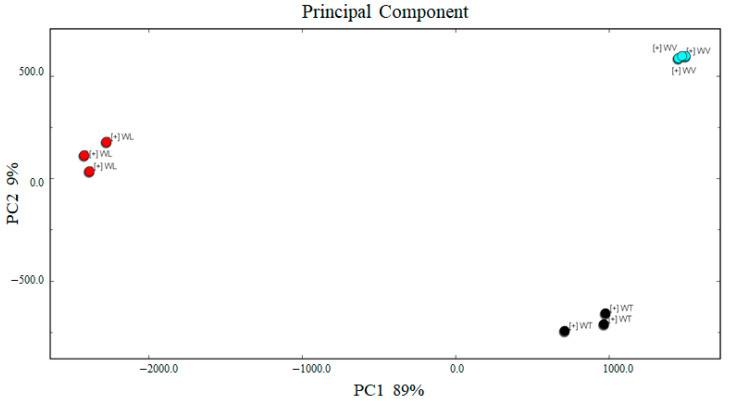
The scores plot of PCA analysis in cherry wines derived from three cultivars.

**Table 1 molecules-27-09056-t001:** Analysis conditions for cherry fruits and their wines by HS-GC-IMS.

Gas Phase-Ion Mobility Spectrometry Unit	
Analysis time	30 min
Column type	MXT-5, 15 m, 0.53 mm ID, 1 μm FT
Column temperature	60 °C
Carrier gas/drift gas	N_2_
IMS temperature	45 °C
**Automatic Headspace Sampling Unit**	
Injection volume	500 μL
Incubation time	20 min
Incubation temperature	40 °C
Syringe temperature	85 °C
Incubation speed	500 rpm

**Table 2 molecules-27-09056-t002:** GC-IMS integration parameters of volatile compounds detected in cherry fruits and their wines.

No.	Compounds	CAS	Formula	MW	RI	RT [sec]	DT [a.u.]
1	Propionic acid	C79094	C_3_H_6_O_2_	74.1	1705.6	1534.95	1.10999
2	Acetic acid	C64197	C_2_H_4_O_2_	60.1	1511.2	1181.04	1.1571
3	Ethyl octanoate-M	C106321	C_10_H_20_O_2_	172.3	1455.8	1080.20	1.48292
4	Ethyl octanoate-D	C106321	C_10_H_20_O_2_	172.3	1455.8	1080.20	2.02662
5	1-Hexanol-M	C111273	C_6_H_14_O	102.2	1361.3	908.19	1.32393
6	1-Hexanol-D	C111273	C_6_H_14_O	102.2	1360.9	907.54	1.64526
7	3-Methyl-1-butanol-M	C123513	C_5_H_12_O	88.1	1212.9	673.05	1.24442
8	3-Methyl-1-butanol-D	C123513	C_5_H_12_O	88.1	1210.4	669.48	1.50227
9	Ethyl hexanoate	C123660	C_8_H_16_O_2_	144.2	1238.5	708.71	1.79654
10	Hexyl acetate-M	C142927	C_8_H_16_O_2_	144.2	1281.7	769.01	1.3894
11	Hexyl acetate-D	C142927	C_8_H_16_O_2_	144.2	1277.8	763.52	1.89309
12	*(E)*-2-hexenal-M	C6728263	C_6_H_10_O	98.1	1225.9	691.14	1.18288
13	*(E)*-2-hexenal-D	C6728263	C_6_H_10_O	98.1	1234.8	703.62	1.51499
14	3-Methyl-3-buten-1-ol	C763326	C_5_H_10_O	86.1	1253.6	729.77	1.17268
15	3-Hydroxy-2-butanone-M	C513860	C_4_H_8_O_2_	88.1	1291.3	782.35	1.06755
16	3-Hydroxy-2-butanone-D	C513860	C_4_H_8_O_2_	88.1	1291.7	782.92	1.33107
17	Benzaldehyde-M	C100527	C_7_H_6_O	106.1	1577	1300.77	1.14883
18	Butanol-M	C71363	C_4_H_10_O	74.1	1146.3	564.09	1.18197
19	Butanol-D	C71363	C_4_H_10_O	74.1	1147.8	566.62	1.38972
20	Isoamyl acetate-M	C123922	C_7_H_14_O_2_	130.2	1128.4	532.65	1.30544
21	Isoamyl acetate-D	C123922	C_7_H_14_O_2_	130.2	1127.9	531.92	1.74943
22	2-Methyl-1-propanol-M	C78831	C_4_H_10_O	74.1	1099.6	482.38	1.17133
23	2-Methyl-1-propanol-D	C78831	C_4_H_10_O	74.1	1098.3	480.17	1.37408
24	1-Propanol	C71238	C_3_H_8_O	60.1	1039.7	413.82	1.26147
25	Isobutyl acetate	C110190	C_6_H_12_O_2_	116.2	1015.7	387.03	1.61493
26	Ethyl propionate	C105373	C_5_H_10_O_2_	102.1	958.9	339.85	1.44949
27	Propyl acetate	C109604	C_5_H_10_O_2_	102.1	951	334.50	1.47663
28	Hexanal-M	C66251	C_6_H_12_O	100.2	1090.6	470.71	1.26538
29	Hexanal-D	C66251	C_6_H_12_O	100.2	1090.7	470.79	1.55908
30	Ethanol	C64175	C_2_H_6_O	46.1	927.5	318.59	1.13044
31	1-Penten-3-ol	C616251	C_5_H_10_O	86.1	1163.2	593.48	0.94008
32	Heptanal	C111717	C_7_H_14_O	114.2	1190.1	640.59	1.33915
33	Ethyl acetate	C141786	C_4_H_8_O_2_	88.1	892.7	295.00	1.33551
34	Methyl acetate	C79209	C_3_H_6_O_2_	74.1	855.8	270.00	1.19134
35	Butanal	C123728	C_4_H_8_O	72.1	837	257.28	1.28121
36	2-Methylbutanal	C96173	C_5_H_10_O	86.1	918.3	312.37	1.39874
37	Propionaldehyde	C123386	C_3_H_6_O	58.1	831.9	253.80	1.14343
38	2-Methyl-1-butanol	C137326	C_5_H_12_O	88.1	1212.2	672.02	1.23289
39	Butyl acetate	C123864	C_6_H_12_O_2_	116.2	1075.5	453.80	1.23674
40	Trans-2-methyl-2-butenal-M	C497030	C_5_H_8_O	84.1	1051.5	426.957	1.09627
41	Trans-2-methyl-2-butenal-D	C497030	C_5_H_8_O	84.1	1047.4	422.414	1.33819
42	Butan-2-ol	C78922	C_4_H_10_O	74.1	1024	396.256	1.14746
43	4-Methyl-2-pentanone	C108101	C_6_H_12_O	100.2	1013.7	384.747	1.17867

MW: molecular weight; RI: retention index; RT: retention time; DT: drift time; D: Dimer; M: Monomer.

## Data Availability

Not applicable.

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
