# Peer review of "HS-GC-IMS and PCA to Characterize the Volatile Flavor Compounds in Three Sweet Cherry Cultivars and Their Wines in China"

_molecules, 2022, doi:10.3390/molecules27249056_

Round 1

Reviewer 1 Report

The manuscript "HS-GC-IMS and PCA to Characterization of the Volatile Flavor Compounds in Three Sweet Cherry Cultivars and Their Wines in China" is interesting and valuable. The characterized differences and sources of volatile flavor compounds in different cherry fruits and their wines were detected. However, there are some points should be considered in order enhancing the research.

1.     Some compounds in cherry wine derived from cherry fruits rather than fermentation process, and authors should particularly explain how to get this conclusion.

2.     The authors have compared the cherry fruits and their wines, I am wondering why don’t compare cherry fruit and wine of same cultivar ? For example, compare Van and WV, you can explain which compound derived from fruit or fermentation process.

3.     The identified kinds of flavor compounds were 27 (fruit) and 23 (wine), why it has less in wines ?

4.     The letters in figures (Figure 2 and Figure 6) are not sharp. Please modify the output version.

5.     The qualitatively characteristic of volatile compounds in different cherry cultivars were represented, and what is the flavor type of different cultivars?

6.     Please uniform name of the compounds, For example, Octanoic acid ethyl ester” to Ethyl octanoate, “Ethyl propanoate” to Ethyl propionate.

Specific points:

L15: Please affirm that whether the “solid phase microextraction” is used or not.

L27: “And other volatile compounds almostly produced by saccharomyces cerevisiae.” should be rewritten.

L63: “such as GC-MS, GC-O.” should add connective, and GC-O needs to be spelled out here.

L71: Please change analyses analyse

L82: The enzymatic activity of pectinex should add

L266: “In conclusion, alcohols, aldehydes and esters were prevalent in cherry fruits, while esters and alcohols in cherry wines.” should be rewritten to draw a distinction between fruit and wine.

Overall, it is a nice work and is possible to be published on Molecules after fully consideration of aforementioned issues.

Author Response

Dear Editors and Reviewers:

Thanks for your efforts in reviewing our manuscript and providing helpful comments and suggestions. We tried our best to improve the manuscript and made some changes in the manuscript. These changes will not influence the content and framework of the paper. We appreciate for Editors and Reviewers’ warm work earnestly, and hope that the correction will meet with approval.

All revisions to the manuscript were marked up using the “Track Changes” function.

Main corrections in the paper and the responses to the reviewer’s comments are as follows:

Response to Reviewer 1 Comments

Point 1: Some compounds in cherry wine derived from cherry fruits rather than fermentation process, and authors should particularly explain how to get this conclusion.

Response 1: Thanks for the reviewer’s suggestions. We found that some compounds showed relatively high content in all three cherry fruits, whereas all aldehydes, two esters (propyl acetate, butyl acetate), organic acids and ketone showed the relatively lower content in cherry wine. Thus, those compounds derived from cherry fruits, which can’t be produced during fermentation.

Point 2: The authors have compared the cherry fruits and their wines, I am wondering why don’t compare cherry fruit and wine of same cultivar ? For example, compare Van and WV, you can explain which compound derived from fruit or fermentation process.

Response 2: Thanks for the reviewer’s question. By comparing the cherry fruits and their wines of every cultivar, we can see that those compounds derived from cherry fruits or fermentation process. It maybe more accurate and strict by summary of different cultivars.

Point 3: The identified kinds of flavor compounds were 27 (fruit) and 23 (wine), why it has less in wines ?

Response 3: Thanks for the reviewer’s question. Although the varieties in wines were less, but the relative content of some volatile flavor compounds (like butanol, 2-methyl-1-propanol, 1-propanol, ethyl octanoate, ethyl hexanoate, isoamyl acetate, isobutyl acetate, ethyl propanoate, propyl acetate and acetic acid) were higher than in fruit. Which showing higher flavor characteristics in wines.

Point 4: The letters in figures (Figure 2 and Figure 6) are not sharp. Please modify the output version.

Response 4: Thanks for the reviewer’s suggestions and we had renewed those figures.

Point 5: The qualitatively characteristic of volatile compounds in different cherry cultivars were represented, and what is the flavor type of different cultivars?

Response 5: Thanks for the reviewer’s question. The aldehydes were the most prevalent in cv. Tie, for example, benzaldehyde, (E)-2-hexenal, hexanal and heptanal. Furthermore, 3-methyl-3-buten-1-ol, 1-penten-3-ol, butan-2-ol and 4-methyl-2-pentanone also showed the most abundant. Almost all esters and alcohols showed the most prevalent in cv. Van. Except for trans-2-methyl-2-butenal and ethyl propanoate, there was no volatile compound that had more prevalent in cv. Lap.

So, compared with cv. Lap, the odor type of cv. Tie were fruity and green, and cv. Van were fruity and ethereal.

Point 6: Please uniform name of the compounds, For example, “Octanoic acid ethyl ester” to Ethyl octanoate, “Ethyl propanoate” to Ethyl propionate.

Response 6: Thanks for the reviewer’s suggestions. The name of the compounds was uniformed.

We changed “Octanoic acid ethyl ester” to Ethyl octanoate, “Ethyl propanoate” to Ethyl propionate. And renewed the figures.

Point 7: L15: Please affirm that whether the “solid phase microextraction” is used or not.

Response 7: Thanks for the reviewer’s suggestions and we deleted ‘solid phase microextraction’.

Point 8: L27: “And other volatile compounds almostly produced by saccharomyces cerevisiae.” should be rewritten.

Response 8: Thanks for the reviewer’s suggestions and we rewrote the sentence:

And other volatile compounds almost produced by saccharomyces cerevisiae.

Point 9: L63: such as GC-MS, GC-O.’ should add connective, and GC-O needs to be spelled out here.

Response 9: Thanks for the reviewer’s suggestions and we change “such as GC-MS, GC-O.” to such as GC-MS, GC-O (Gas chromatography-olfactometry) and coupled with other means.

Point 10: L71: Please change analyses → analyse

Response 10: Thanks for the reviewer’s suggestion and we change “analyses” to analyse.

Point 11: L82: The enzymatic activity of pectinex should add

Response 11: Thanks for the reviewer’s suggestion and we added the enzymatic activity of pectinex (1 000 U/g).

Point 12: L266: ‘In conclusion, alcohols, aldehydes and esters were prevalent in cherry fruits, while esters and alcohols in cherry wines.’ should be rewritten to draw a distinction between fruit and wine.

Response 12: Thanks for the reviewer’s suggestion. And we rewrote the sentence:

In conclusion, aldehydes, alcohols and esters were prevalent in cherry fruits, and huge diversity in content of aldehydes. while esters and alcohols were ubiquitous in wines.

Please refer to details in the revised manuscript.

The above is our revised responses based on suggestions of the editor and reviewers.

Once again, thank you very much for your comments and suggestions.

We look forward to your reply!

Sincerely,

Prof. Na Zhang

Harbin University of Commerce

Reviewer 2 Report

The aim of this research was to chemically characterise the volatile profile of sweet cherry fruits and their respective wines by means HS-GC-IMS. It is an interesting manuscript, that could be eventually published if it will be extensively revised.

To improve the manuscript, I would suggest the following:

1)    Revise English, many sentences are difficult to understand (70-72, 76-77 etc) and there are several grammar (line 60-61, line 103, line 115-116, line 121-122 etc) or typing errors.

2)    Introduction should be improved. The introduction section lacks information on the analytical side of VOCs detection, identification and characterisation. Authors write that ‘there are several ways for measuring volatile compounds by using various techniques, such as GC-MS and GC-O’ but they do not include any references. Finally, the authors should point out in the introduction that this study is an exploratory research, given also, the low number of investigated samples. Therefore, it is not possible to perform  ‘cluster and similarity analysis’ as it is written by the authors in line 108 (this line should be deleted).

3)    Line 69-70. What does the sentence ‘PCA is a key technique to distinguish the differences of multiple samples by multiple software’ mean? Please better clarify it.

4) In my opinion, authors should deeply revise statistical analysis. In section 2.4 Statistical Analysis, authors write to use PCA, but no information is given on the nature of the analysed data (did authors consider the area of the peak? or the volume of 2D peaks?) on the dimension of the analysed matrices (how many raw and columns?), on the used pretreatment (mean centering? Autoscaling?). Please, clarify this point for each analysed data set (cherry and wines). Concerning the comments of PCA results, authors have to better describe Figure 4 and Figure 8 (i.e. the scores plot of PCA analysis carried out on cherry and wine data, respectively) and to argument the differences among samples considering the respective loadings plot.

5) Lastly, I suggest naming the graphs, Figure 4 and Figure 8 as scores plot and to report the respective loadings plot as well.

Author Response

Dear Editors and Reviewers:

Thanks for your efforts in reviewing our manuscript and providing helpful comments and suggestions. We tried our best to improve the manuscript and made some changes in the manuscript. These changes will not influence the content and framework of the paper. We appreciate for Editors and Reviewers’ warm work earnestly, and hope that the correction will meet with approval.

All revisions to the manuscript were marked up using the “Track Changes” function.

Main corrections in the paper and the responses to the reviewer’s comments are as follows:

Response to Reviewer 2 Comments

Point 1: Revise English, many sentences are difficult to understand (70-72, 76-77 etc) and there are several grammar (line 60-61, line 103, line 115-116, line 121-122 etc) or typing errors.

Response 1: Thanks for the reviewer’s suggestions. We rewrote those sentences and revised the full text.

The details could be seen in the revised manuscript.

Point 2: Introduction should be improved. The introduction section lacks information on the analytical side of VOCs detection, identification and characterisation. Authors write that ‘there are several ways for measuring volatile compounds by using various techniques, such as GC-MS and GC-O’ but they do not include any references. Finally, the authors should point out in the introduction that this study is an exploratory research, given also, the low number of investigated samples. Therefore, it is not possible to perform ‘cluster and similarity analysis’ as it is written by the authors in line 108 (this line should be deleted).

Response 2: Thanks for the reviewer’s suggestions. We rewrote those sentences:

This far, there are several ways for measuring volatile compounds by using various techniques to enhance and distinguish the characteristics of cherry wines, such as GC-MS, GC-O (Gas chromatography-olfactometry) and coupled with other means [16,19,20]. GC-MS is most applied tool as to differentiate cherry and wines in fruit development, raw materials, cultures and processing methods [18-21].

PCA is a key technique to distinguish the differences of multiple samples by summa-rizing into much fewer variables called scores which are weighted average of the original variables.

As an exploratory research, this work may promote application and quality improve-ment for making of cherry alcohol beverage.

Point 3: Line 69-70. What does the sentence ‘PCA is a key technique to distinguish the differences of multiple samples by multiple software’ mean? Please better clarify it.

Response 3: Thanks very much for questions. We rewrote the sentence:

PCA is a key technique to distinguish the differences of multiple samples by summarizing into much fewer variables called scores which are weighted average of the original variables.

Point 4: In my opinion, authors should deeply revise statistical analysis. In section 2.4 Statistical Analysis, authors write to use PCA, but no information is given on the nature of the analysed data (did authors consider the area of the peak? or the volume of 2D peaks?) on the dimension of the analysed matrices (how many raw and columns?), on the used pretreatment (mean centering? Autoscaling?). Please, clarify this point for each analysed data set (cherry and wines). Concerning the comments of PCA results, authors have to better describe Figure 4 and Figure 8 (i.e. the scores plot of PCA analysis carried out on cherry and wine data, respectively) and to argument the differences among samples considering the respective loadings plot.

Response 4: Thanks for the reviewer’s suggestions.

The analysed data:

PCA was performed with the peak position and intensity of the volatile compound by the dynamic PCA plugin to ascertain the differences among tested samples.

We renamed Figure 4 and Figure 8.

The contribution of volatile compounds to the differentiation was added:

The main contributing volatile compounds of cv. Tie to the differentiation were 4-methyl-2-pentanone heptanal, (E)-2-hexenal, 3-methyl-3-buten-1-ol, benzaldehyde, butan-2-ol, ethyl acetate, methyl acetate. 2-methyl-1-propanol, trans-2-methyl-2-butenal, ethyl propionate, 3-methyl-1-butanol and acetic acid contributed to cv. Lap. Hexyl ace-tate, isoamyl acetate, 3-methyl-1-butanol, 2-methyl-1-propanol, 3-hydroxy-2-butanone, propionaldehyde, isobutyl acetate, 1-hexanol and butanal contributed to cv. Van. Hence compared with cv. Lap, the odor type of cv. Tie were fruity and green, and cv. Van were fruity and ethereal.

The main contributing volatile compounds of WT to the differentiation were propyl ac-etate, 1-penten-3-ol, 2-methyl-1-propanol and 1-hexanol. Isobutyl acetate, isoamyl acetate, ethyl propionate, ethyl acetate, methyl acetate, ethyl hexanoate and ethyl octanoate contributed to WL. Butanol, butyl acetate, 1-propanol, butanol and acetic acid contributed to WV. Those correspond to the result of Gallery Plot.

The loading plot was added as Supplementary figures.

Point 5: Lastly, I suggest naming the graphs, Figure 4 and Figure 8 as scores plot and to report the respective loadings plot as well

Response 5: Thanks for the reviewer’s suggestions. We renamed Figure 4 and Figure 8 and added loading plot was added as Supplementary figures.

Please refer to details in the revised manuscript.

The above is our revised responses based on suggestions of the editor and reviewers.

Once again, thank you very much for your comments and suggestions.

We look forward to your reply!

Sincerely,

Prof. Na Zhang

Harbin University of Commerce

Round 2

Reviewer 2 Report

The authors partially answered to my comments. No information has been provided on the data pretreatment used  in the PCA yet.  Was data autoscaled or mean centered? In the data set used for PCA, authors used 9 samples (3 samples X 3 replicates). On the other side, it is not clear how many variables have been used. Please, clarify both these aspects (used pretreatments and number of used variables) in the manuscript.

Figure S1 and S2 have been reported as supplementary materials, but they are never mentioned in the text. 

The quality of Figure S1 and S2 should be improved. The name of used variables is not readable in both figures. Furthermore, the reported figures are not loading plots but PCA biplots, since both scores and loadings have been reported.  The correct acronym is PCA and not PAC. Please correct the captions of Figure S1 and S2. The origin point (the point 0,0) reported in Figure S1 is not clear. Indeed, it does not seem to have PC1 equal to zero but a negative value. 

Finally, the sentence 'by summarizing into much fewer variables called scores which are weighted average of the original variables' (line 77-78) is not correct. The latent variables or principal components are the weighted average of the original variables and not the scores. Please correct the sentence. 

Author Response

Dear Editors and Reviewers:

We sincerely thank the editor and reviewer for their valuable feedback that we have used to improve the quality of our manuscript. All revisions to the manuscript were marked up using the “Track Changes” function.

According to your nice suggestions, we have made extensive corrections to our previous draft, the detailed corrections are listed below.

Response to Reviewer 2 Comments

Point 1: The authors partially answered to my comments. No information has been provided on the data pretreatment used in the PCA yet. Was data autoscaled or mean centered? In the data set used for PCA, authors used 9 samples (3 samples X 3 replicates). On the other side, it is not clear how many variables have been used. Please, clarify both these aspects (used pretreatments and number of used variables) in the manuscript.

Response 1: Thanks for the reviewer’s suggestions. We rewrote some sentences in statistical analysis:

The spectrogram was analysed by its accompanied software, including VOCal and three plugins (Reporter, Gallery Plot and Dynamic PCA plugins).

The data were processed with the peak intensity of detected volatile compounds by using dynamic PCA plugin. The PCA data matrix of cherry fruits had 42 columns (names of volatile compounds) and nine rows (three samples with three replicates). In cherry wines, there were 35 columns and nine rows. Then, the peak intensity was scaled and mapped to appropriate intervals from the matrix automatically. The principal components were determined and the corresponding contribution rate calculations were performed. Finally, scores and loading plots were drawn to detect the regularity and difference of cherry fruits and their wines.

Point 2: Figure S1 and S2 have been reported as supplementary materials, but they are never mentioned in the text.

Response 2: Thanks for the reviewer’s suggestion. We added supplementary materials in the manuscript.

Point 3: The quality of Figure S1 and S2 should be improved. The name of used variables is not readable in both figures. Furthermore, the reported figures are not loading plots but PCA biplots, since both scores and loadings have been reported. The correct acronym is PCA and not PAC. Please correct the captions of Figure S1 and S2. The origin point (the point 0,0) reported in Figure S1 is not clear. Indeed, it does not seem to have PC1 equal to zero but a negative value.

Response 3: Thanks very much for suggestions. We feel sorry for our carelessness. We renew the loading plots figures and captions.

The dynamic PCA plugin (a accompanied plugin in instrument) could only export those loading plots figures, so we put the figures in supplementary materials instead of manuscript.

Point 4: Finally, the sentence 'by summarizing into much fewer variables called scores which are weighted average of the original variables' (line 77-78) is not correct. The latent variables or principal components are the weighted average of the original variables and not the scores. Please correct the sentence.

Response 4: Thanks for the reviewer’s suggestions. We corrected the sentence:

‘by summarizing into much fewer variables which are weighted average of the original variables.’

Please refer to details in the revised manuscript.

We tried our best to improve the manuscript. We appreciate for Editors and Reviewers’ warm work earnestly, and hope the correction will meet with approval. Once again, thank you very much for your comments and suggestions.

We look forward to your reply!

Sincerely,

Prof. Na Zhang

Harbin University of Commerce
